# Hierarchical Neural Simulation-Based Inference Over Event Ensembles

**Lukas Heinrich**                                        *l.heinrich@tum.de*
*Technical University of Munich*

**Siddharth Mishra-Sharma**                               *smsharma@mit.edu*
*MIT, Harvard University, IAIFI*

**Chris Pollard**                            *christopher.pollard@warwick.ac.uk*
*University of Warwick*

**Philipp Windischhofer**                          *windischhofer@uchicago.edu*
*University of Chicago*

**Reviewed on OpenReview:** *https://openreview.net/forum?id=Jy2IgzjoFH*

## Abstract

When analyzing real-world data it is common to work with event ensembles, which comprise sets of observations that collectively constrain the parameters of an underlying model of interest. Such models often have a hierarchical structure, where "local" parameters impact individual events and "global" parameters influence the entire dataset. We introduce practical approaches for frequentist and Bayesian dataset-wide probabilistic inference in cases where the likelihood is intractable, but simulations can be realized via a hierarchical forward model. We construct neural estimators for the likelihood(-ratio) or posterior and show that explicitly accounting for the model's hierarchical structure can lead to significantly tighter parameter constraints. We ground our discussion using case studies from the physical sciences, focusing on examples from particle physics and cosmology.

## 1 Introduction

Datasets composed of multiple samples are ubiquitous in scientific and more broadly real-world data analysis tasks. These datasets are typically governed by underlying models that exhibit a rich hierarchical structure, with local parameters shaping individual events while global parameters exert influence across the entire dataset. This layered structure, if appropriately utilized, can greatly augment the efficiency and effectiveness of the inference process.

The complexity of scientific models and high-dimensionality of datasets has led to a recent surge in interest in implicitly-specified models, where the likelihood function is intractable but simulations can be realized via mechanistic forward modeling. The paradigm of simulation-based inference (SBI), augmented using tools from machine learning and differentiable optimization more broadly, has emerged as a powerful approach for performing inference in such scenarios (Cranmer et al., 2020). However, the majority of existing methods for simulation-based inference are designed for learning from individual data points and do not fully capitalize on the hierarchical structure of the data-generating process.

To address these gaps, we propose a set of novel approaches that augment existing simulation-based inference techniques, in both frequentist and Bayesian paradigms, with the goal of exploiting the hierarchical structure of the governing models. We contextualize our discussion using case studies from the physical sciences, with a particular focus on particle physics (particle collider data) and astrophysics (strong gravitational lensing images).

In the present work we make several methodological contributions that are necessary for optimal deployment of simulation-based inference methods on large-scale scientific datasets. Our primary contributions are the following:

- We substantiate theoretically as well as empirically the fact that hierarchy-aware inference in many implicit models with a hierarchical structure requires a dataset-wide approach, contrasted with the more common paradigm of combining implicit likelihood or posterior estimators associated with individual observations. By "hierarchy-aware inference", we refer to building a *still approximate* posterior or likelihood-ratio estimator that nevertheless takes into account the full (in our case) hierarchical structure of the assumed forward model. We systematically derive conditions under which such a hierarchical approach is necessary for correct inference, depending on how the model parameters are partitioned into local and global parameters of interest and nuisance parameters;

- We introduce frequentist as well as Bayesian methods for dataset-wide learning that can be used to perform simulation-based inference over event ensembles and can deal with datasets of varying cardinality. We connect our insights to several common use cases in the physical sciences and show how popular simulation-based inference paradigms can be adapted for optimal dataset-wide learning. In particular, we introduce the first approach for end-to-end frequentist simulation-based inference targeting hierarchical set-valued data;

- We show that our machine learning-based inference methods are generically substantially faster than traditional approaches, such as Markov Chain Monte Carlo (MCMC) methods, even when the likelihood is tractable, while giving consistent results. They also allow performing inference in "streaming" mode where, e.g., posterior estimates are efficiently updated in real-time as new observations are made without having to perform a re-analysis of the entire updated dataset.

The paper is organized as follows. In Sec. 2 we provide a background discussion of hierarchical frequentist as well as Bayesian inference, and describe the class of forward models we target. We discuss related work in context of this background Sec. 3. In Sec. 4 we describe the methods and architectures used in this paper. Section 5 presents several case studies including a simple toy example, a particle physics example, and an example from astrophysics, in increasing order of complexity. We conclude in Sec. 6.

## 2 Theory and background

We use the following notation throughout: $x_i$ refers to an observation associated with an individual event, $\theta$ are the global (dataset-wide) parameters, and $z_i$ are the local (per-event) parameters. Ensembles of either observations or variables are denoted with curly braces, $\{x\} \equiv \{x\}_{i=1}^{N}$. We will also differentiate nuisance parameters, which are parameters associated with the data-generating process that we are generally not interested in inferring; these will be treated through either profiling (defined later) or marginalization. Nuisance parameters will be denoted by $\nu$, or $\theta_\nu$ and $z_\nu$ when we wish to distinguish between global and local nuisance parameters.

### 2.1 Hierarchical models

The joint probability distribution of a set of events with cardinality $N$, with global parameters of interest $\theta$, global nuisance parameters $\theta_\nu$ as well as local parameters $z_i$ can be written as:

$$p(\{x\} \mid \{z\}, \theta, \theta_\nu) = \prod_{i=1}^{N} p\left(x_i \mid z_i, \theta, \theta_\nu\right), \tag{1}$$

In many scientific applications, the number of events in a dataset may not be known a priori. This could occur if events are observed sequentially (such as gravitational wave events observed over time), if we aim to apply the same model to datasets with different event counts (e.g., astronomical observations of sky patches containing different numbers of stars/galaxies), or if the rate of events is itself a model parameter that may

be interesting to measure (e.g., an interaction cross section at a particle collider experiment). In these cases, we can generalize the joint probability to an *extended model*:

$$p(\{x\} \mid \{z\}, \theta, \theta_\nu) = \sum_{N=0}^{\infty} p(N \mid \theta) \prod_{i=1}^{N} p(x_i \mid z_i, \theta, \theta_\nu). \tag{2}$$

Here, $p(N \mid \theta)$ represents the probability of observing $N$ events given the global parameters $\theta$. Often, $p(N \mid \theta)$ is a Poisson distribution, and the full model is known as a *marked Poisson process*. Our proposed techniques apply to both situations, and we present example applications described by equation 1 as well as equation 2 below.

## 2.2 Bayesian inference

In the Bayesian paradigm, we introduce a prior $p(\{z\}, \theta, \theta_\nu) = p(\theta, \theta_\nu) \prod_i p(z_i \mid \theta, \theta_\nu)$ and wish to target the posterior distribution over the global as well as local parameters of interest (and marginalizations thereof) given a dataset.

In special cases, it is possible to combine event-level inferences to construct the dataset-wide inference. For example, in the non-extended case one can construct the dataset-wide *full* posterior from the event-level posteriors $p(\theta, \theta_\nu, z_i \mid x_i)$,

$$p(\theta, \theta_\nu, \{z\} \mid \{x\}) = \frac{1}{p(\theta, \theta_\nu)^{N-1}} \prod_{i=1}^{N} p(\theta, \theta_\nu, z_i \mid x_i), \tag{3}$$

where the prior factor in the denominator ensures that the prior density is not counted multiple times. Similarly, if one has access to the per-event posteriors marginalized over local parameters $z_i$, and the target quantity is the dataset-wide posterior *marginalized* over local nuisance parameters, a combination is possible due to the independence of the events and the assumption that the various priors also factorize:

$$p(\theta \mid \{x\}) = \int \mathrm{d}\{z_\nu\} \, p(\theta, \{z_\nu\} \mid \{x\}) = \int \mathrm{d}\{z_\nu\} \frac{p(\{x\} \mid \theta, \{z_\nu\}) \cdot p(\theta, \{z_\nu\})}{p(\{x\})}$$
$$= \left[ \prod_i \int \mathrm{d}z_{\nu,i} \frac{p(x_i \mid \theta, z_{\nu,i})}{p(x_i)} \cdot p(z_{\nu,i} \mid \theta) \right] p(\theta) = \frac{1}{p(\theta)^{N-1}} \prod_{i=1}^{N} p(\theta \mid x_i) \tag{4}$$

However in the general case such combinations do not hold; e.g., it is not possible to combine marginal per-event posteriors over *global* nuisance parameters $p(\theta \mid x_i) = \int \mathrm{d}\theta_\nu \, p(\theta, \theta_\nu \mid x_i)$ to construct the dataset-wide posterior $p(\theta \mid \{x\})$, marginalized over global and local nuisance parameters. Therefore, a general solution for marginalized dataset-wide posteriors *necessitates* a dataset-wide approach to inference. This can be done either by inferring global parameters at a dataset-wide level (in particular if the dimensionality of the resulting posterior is sufficiently wieldy), or by performing dataset-level marginalization.

## 2.3 Frequentist inference

In the frequentist setting, inference on parameters is typically split into procedures for point estimation and interval estimation. A popular choice for a point estimator is the maximum-likelihood estimator (MLE), due to its favorable asymptotic property of being the minimum-variance unbiased estimator. Interval estimation and hypothesis testing are based on a subjective choice of a test statistic $t_{\theta,\theta_\nu}(x)$ and a desired confidence level:

$$\hat{\theta}, \hat{\theta}_\nu = \mathrm{argmin}_{\theta,\theta_\nu} \left[ -\log L_{\{x\}}(\theta, \theta_\nu) \right]; \qquad I_{\theta,\theta_\nu}^\alpha = \{\theta \mid t_{\theta,\theta_\nu}(\{x\}) < t_{\theta,\theta_\nu}^\alpha\} \tag{5}$$

where $L_{\{x\}}(\theta, \theta_\nu)$ is a function proportional to the likelihood $p(\{x\} \mid \theta, \theta_\nu)$ and $t_{\theta,\theta_\nu}^\alpha$ is the value of the quantile function at $1 - \alpha$ cumulative probability and serves as the threshold value for the interval. Here, $\alpha$ indicates the *size* of a hypothesis test, with a typical value being $\alpha = 0.05$.

As in the Bayesian case, one is often interested in an inference that only considers the parameters of interest $\theta$ and removes any dependence on nuisance parameters $\theta_\nu$. The frequentist analogue to (partial) marginalization

is (partial) optimization (also referred to as "profiling"). For interval estimation and hypothesis testing the test statistic is thus modified to read

$$t_\theta(\{x\}) = -\log \frac{p(\{x\} \mid \theta, \hat{\hat{\theta}}_\nu)}{p(\{x\} \mid \hat{\theta}, \hat{\theta}_\nu)}. \tag{6}$$

where $\hat{\hat{\theta}}_\nu = \hat{\theta}_\nu(\{x\}, \theta)$ is the optimal value for $\theta_\nu$ when $\theta$ is fixed to a particular value. Similarly to the Bayesian case, it is in general not possible to reconstruct the data-set wide nuisance-free inferences from per-event nuisance free inferences: In general the dataset-wide profile likelihood ratios cannot be constructed from per-event profile likelihood ratios.

### 2.4 Simulation-based inference

When the likelihood is not tractable but simulations can be realized via forward modeling, neural simulation-based inference methods can be leveraged to perform inference. In cases where event-level inference quantities can be combined to yield a correct dataset-wide quantity, neural posterior or likelihood-ratio estimators trained on individual events can be combined. For example, marginal posteriors $p(\theta, \theta_\nu \mid x_i)$ can be trained and combined using equation 4.

Alternatively, a per-event likelihood ratio estimator parameterized by all parameters, including nuisance parameters, can be learned (Cranmer et al., 2015; Nachman & Thaler, 2021) to train a neural estimate for the per-event likelihood ratio $s_\phi(x_i, \theta, \theta_\nu) = \frac{p(x_i|\theta,\theta_\nu)}{p(x_i|\theta_0,\theta_{\nu,0})}$, These can be used to build a dataset-wide likelihood ratio $\prod_i \frac{p(x_i|\theta,\theta_\nu)}{p(x_i|\theta_0,\theta_{\nu,0})}$ that can be an input for a dataset-wide statistical analysis to yield posteriors or likelihood ratios which can then be marginalized or profiled over to yield dataset-level quantities. While this simplifies the training procedure as the networks are only performing inference for single events, a disadvantage of this approach is that the downstream statistical treatment can be very costly if the number of nuisance parameters $\theta_\nu$ is large, so that the amortization gains from the neural estimates are not fully exploited.

Given these nuances, methods for set-wide simulation-based inference necessitate architectures which can efficiently model joint posteriors or likelihood ratio over global and local parameters of interest, for sets of varying cardinality, without requiring explicit parameterization over a potentially large number of nuisance parameters (which would otherwise be necessary for hierarchy-aware inference with event-level estimators). In the following, we describe two such architectures for amortized dataset-wide inferences.

## 3   Related work

The methods presented in this paper build upon previous studies across several disciplines. A likelihood-free algorithm for obtaining frequentist profile likelihood ratios corresponding to *individual* observations was previously presented in Heinrich (2022). Also in the context of frequentist inference, Nachman & Thaler (2021) explored dataset-wide inference in the context of particle collider studies, without distinguishing nuisance parameters and parameters of interest.

Hierarchical inference for implicit models has been studied using Gaussian posterior density estimators in Wagner-Carena et al. (2021; 2022), where again individual (per-event) posteriors are learned and combined to obtain estimates for the global parameters. Agrawal & Domke (2021) considered amortized variational inference for a simple class of hierarchical models. While similar in spirit, we consider hierarchical inference in the language of modern scientific simulation-based inference methods, and discuss subtleties associated with treating classes of variables in different hierarchies (local and global) as nuisance parameters.

Hierarchical Neural Posterior Estimation (Rodrigues et al., 2021) is a recently-introduced simulation-based inference method closely related to the deep set-based Bayesian inference model used here. We complement this approach by introducing frequentist analogues for dataset-wide learning; by including amortized estimators that can then target datasets of varying sizes; and by comparing the proposed approach to more traditional methods in terms of inference computational time. Finally, Geffner et al. (2022) introduced a method for efficient set-wide learning by combining score estimators targeting a sub-set of events. Our method is

complementary, focusing on the unexplored setting of hierarchical models and bridging applications to domain sciences.

## 4 Methods

Our goal is to target the graphical models described by equation 1 and equation 2 in a simulation-based inference setting. We wish to construct amortized estimators for the likelihood ratio or posterior corresponding to these models such that they *(1)* are sensitive to the full hierarchical structure of the forward model – i.e., take a *dataset-wide* view, treating local and global parameters separately, *(2)* distinguish between nuisance parameters and parameters of interest at various hierarchy levels, and *(3)* can process datasets of varying cardinality. While previous related works, discussed in Sec. 3, have considered subsets of these requirements, our goal is to target all three with a particular eye towards scientific applications where event ensembles are common. We consider the following architectures, which satisfy these desiderata.

**Deep set-based model**  Here we consider a simple deep sets-inspired (Zaheer et al., 2017) architecture, where per-event encoded features $s_\phi^{\text{glob}}(x_i)$ are aggregated via a permutation-invariant pooling and passed through a decoder network $g_\psi$ that outputs the parameters of a variational posterior distribution for the global parameters $q_\varphi^{\text{glob}}(\theta \mid \{x\})$ (Papamakarios & Murray, 2016). At the same time, local per-event embeddings $s_\phi^{\text{loc}}(x_i)$ are used to condition a density estimator for the local parameters, $q_{\varphi'}^{\text{loc}}(z \mid \{x\}, \theta)$, including a conditional dependence on the global parameter $\theta$. In practice, the local and global embeddings are obtained by chunking a feature vector obtained with $s_\phi$. For the variational ansatz we consider either a multivariate normal distribution or a normalizing flow (Rezende & Mohamed, 2015), with the decoder outputting either the mean-covariance of the normal distribution or the conditioning context for a normalizing flow. The (negative) sum of the local and global posterior log-densities is used as the minimization objective,

$$-\mathcal{L} = \log q_\varphi^{\text{glob}}\left(\theta \mid g_\psi\left(\sum_{i=1}^{N} s_\phi^{\text{glob}}(x_i)\right)\right) + \sum_{i=1}^{N} \log q_{\varphi'}^{\text{loc}}\left(z_i \mid s_\phi^{\text{loc}}(x_i), \theta\right). \tag{7}$$

We derive and further motivate this loss function in App. B. For models of the form of equation 2, the cardinality of the input set is drawn as $N \sim p(N \mid \theta)$ in the training. The density estimator parameters $\{\varphi, \varphi'\}$ as well as the embedding network and decoder parameters $\phi$ and $\psi$ are optimized simultaneously. We note that this is similar to the set-up used in Rodrigues et al. (2021). Fig. 1 shows a schematic illustration of this deep set-based architecture.

**Transformer-based model**  As an alternative to deep set-based aggregation with varying cardinality at training time, we consider a decoder-only transformer (Vaswani et al., 2017; Phuong & Hutter, 2022) which takes feature embeddings from individual events and produces cumulative posterior embeddings using blocks of self-attention and dense layers, applying a causal mask to ensure that output embeddings are only sensitive to preceding events in the sequence. The embeddings are then used to predict the posterior mean and covariance of the variational Gaussian ansatz. A potential advantage of the transformer approach is that, assuming a uniform distribution on the cardinality $N$, we do not need to vary the cardinality of the input set during training. Since the loss consists of a sum of log-likelihoods corresponding to cumulative posteriors, one for each cardinality $N$ up to some specified maximum value, all cardinalities are considered together; $-\mathcal{L} = \sum_N \log q_\varphi^{\text{glob}}(\theta \mid \{x\}_{i=1}^N) + \sum_i \log q_{\varphi'}^{\text{loc}}(z_i \mid x_i, \theta)$, leaving the data-point encodings implicit.

Networks that used LSTM cells to iteratively update a posterior embedding based on incoming data embeddings were also considered, but their convergence properties were noticeably poorer than those of the deep set- and transformer-inspired architectures described above.

## 5 Experiments

We describe several case studies of dataset-wide learning, ranging from illustrative "toy" experiments to more prototypical examples representing problems in particle physics and astrophysics. We refer to App. A for

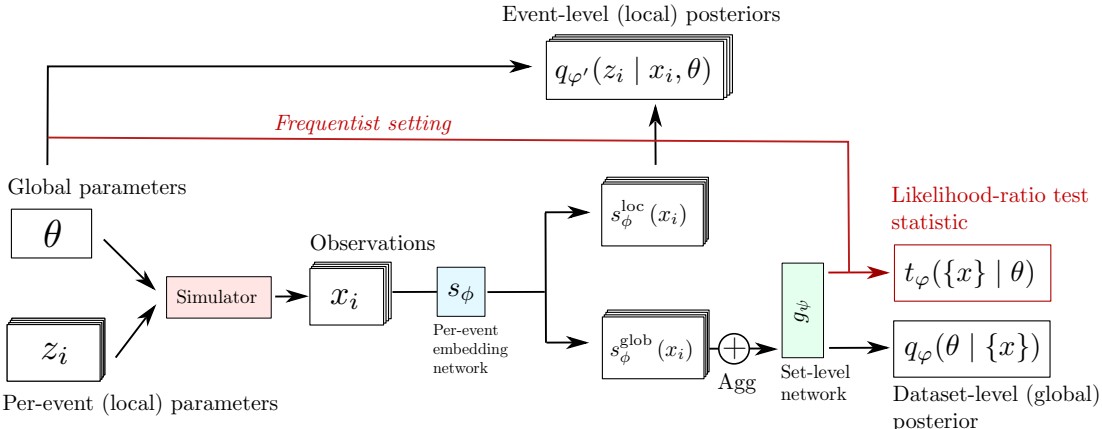

Figure 1: Schematic illustration of the deep set-based architecture used in this work. The red lines/arrows show the path used only in the frequentist setting for training a global test-statistic estimator while profiling over global nuisance parameters.

additional details on the experiments, including details on training. All experiments are implemented using `PyTorch` (Paszke et al., 2019).

### 5.1 Simple multi-variate normal likelihood

**The forward model** To verify the ability of our method to recover the true posterior distribution for sets of varying cardinality, we consider a simple multivariate normal likelihood with known covariance matrix $\Sigma$; $p(\{x\} \mid \theta, \Sigma) = \prod_{i=0}^{N} \mathrm{No}(x_i \mid \theta, \Sigma)$.

The model parameter is the mean vector $\theta$, and $\mu_0, \Sigma_0$ are the hyperparameters of the prior multivariate normal distribution; $p(\theta) = \mathrm{No}(\mu_0, \Sigma_0)$. In this case, the posterior distribution $p(\theta \mid \{x\})$ is also a multivariate normal – the prior and posterior are conjugate – with mean $\mu_{\mathrm{post}}$ and covariance $\Sigma_{\mathrm{post}}$ of an updated posterior given by $\mu_{\mathrm{post}} = \left(\Sigma_0^{-1} + N\Sigma^{-1}\right)^{-1} \left(\Sigma_0^{-1}\mu_0 + N\Sigma^{-1}\overline{x}\right)$ and $\Sigma_{\mathrm{post}} = \left(\Sigma_0^{-1} + N\Sigma^{-1}\right)^{-1}$ where $\overline{x}$ is the sample mean of $x_i$ and $N$ is the total cardinality of the dataset. We choose $\Sigma = \mathrm{diag}(2, 4, 6)$, with each sample consisting of 5 draws from the multi-variate normal distribution; each individual data point then consists of 15 features.

**Inference** We train the deep set and transformer architectures described in Sec. 4 on 50,000 samples drawn from this likelihood with prior $p(\mu) = \mathrm{No}(0, 3)$ and a maximum sequence length of $N_{\mathrm{max}} = 200$. The cardinality of the training set is randomly varying as $N \sim \mathrm{Unif}(1, N_{\mathrm{max}})$. Further details on the training procedure are provided in App. A.

**Results** The model is then tested on 500 new sequences, and the distribution (median and middle-68% containment) over inferred standard deviation $\sigma$ for each of the 3 parameters is shown in Fig. 2 (left: deep set, middle: transformer) compared to the true expected scaling of the parameters (dashed lines). We show $\sigma_1$, $\sigma_2$, and $\sigma_3$, which are the diagonal entries of $\Sigma_{\mathrm{post}}$, the covariance of the posterior on the target mean parameters $\theta$. The deep set model typically gives more faithful posterior widths compared to the transformer, potentially due to the relatively simple nature of the problem combined with the more data-hungry nature of the transformer architecture. Given this, we restrict our experiments in subsequent sections to use the deep set-based architecture, noting that the transformer approach may nevertheless yield advantages depending on the specific structure of the forward model. The right plot shows the evolution of a posterior for a specific sequence using the deep set model, illustrating convergence of the posterior mass around the true point as more data points are included. We note that although this is a simple, analytically tractable example, the final posterior is obtained by analyzing a dataset of dimensionality 3000 – far from a trivial task.

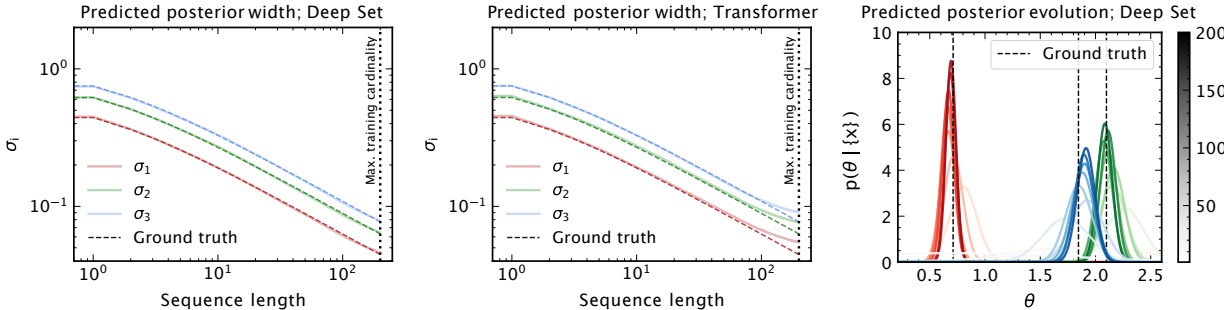

Figure 2: The median and middle-68% containment of the inferred posterior width over 500 test samples as a function of the size of the dataset. Results for the deep sets (left) and transformer (middle) architectures are shown. The expected scaling (dashed lines) is observed in all cases, but the deep set architecture shows a narrower spread in outcomes. The right plot shows the evolution of a posterior for a specific sequence using the deep set model, illustrating convergence of the posterior mass around the true point.

## 5.2 Mixture models in particle physics: frequentist treatment

We demonstrate the dataset level profiling by adapting the approach described in Heinrich (2022) and compare the trained test statistic to the true profile likelihood ratio (not to be confused by the likelihood-to-evidence ratio (Miller et al., 2022)).

**The forward model** In this example we consider a situation that is common in both particle and astrophysics. Here, the observed data can often originates from multiple disparate physics processes of varying intensity (for example a signal process and a background process), such that the overall model is a mixture of the per-process models:

$$p(\{x\} \mid \theta, \theta_\nu) = \text{Pois}(N \mid \lambda(\theta, \theta_\nu)) \prod_{i=1}^{N} (c_s p_s(x_i \mid \theta, \theta_\nu) + (1 - c_s) p_b(x_i \mid \theta, \theta_\nu)); \text{ with,} \tag{8}$$

with $c_s = c_s(\theta, \theta_\nu)$ being the relative signal strength and $p_s$ and $p_b$ the per-event densities for signal and background, respectively. Fast inference for such models is especially important in settings that perform real-time monitoring of the quality of the data stream delivered by the experiment. The observed per-event densities are typically not tractable, but high-fidelity simulators exist. In this work we focus on Gaussian mixture components, since the empirical distributions typically considered in the domain use-cases feature multi-modal densities (e.g., in searches for "bump" features). To demonstrate the simulation-based frequentist inference, we consider the special case where $p_s$ and $p_b$ are Normal distributions $\text{No}(\mu, \sigma)$ with fixed hyperparameters $\mu_{s/b}, \sigma_{s/b}$, but the component coefficients are a function of both the parameter of interest (the "signal strength") $\theta$ as well as the nuisance parameter (i.e. "the background strength"): $c_s = \theta/(\theta + \theta_\nu)$. The overall expected rate is then $\lambda(\theta, \theta_\nu) = \theta + \theta_\nu$.

**Inference** The model is again based on the deep set-inspired architecture described in Sec. 4, with a small variation – after aggregating the per-event embeddings, these are concatenated with the parameters of interest $\theta$ in order to obtain a parameterized neural network classifier one-to-one with the desired test statistic (Cranmer et al., 2015), $s_\phi(\{x\}, \theta) \stackrel{\sim}{\leftrightarrow} t_\theta(\{x\})$. Following the procedure in Heinrich (2022) we aim to train a test statistic with best average power across alternatives irrespective of the nuisance parameters and show that the result is a good approximation of the profile likelihood ratio, i.e., the dataset-wide test statistic typically used for interval estimation and hypothesis testing in frequentist inference:

$$s_\phi(\{x\}, \theta) \stackrel{\sim}{\leftrightarrow} t_\theta(\{x\}) = -2 \log \frac{p(\{x\} \mid \theta, \hat{\hat{\theta}}_\nu)}{p(\{x\} \mid \hat{\theta}, \hat{\theta}_\nu)}. \tag{9}$$

Details on the training procedure as well as hyperparameter choices are provided in App. A.

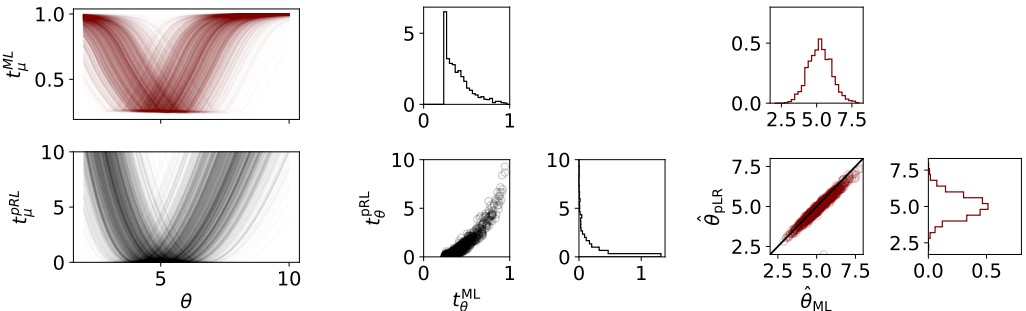

Figure 3: (Left) True profile likelihood ratio for the mixture model scenario in Sec. 5.2 (bottom) compared to the neural test statistic (top). (Center) Neural test statistic and profile likelihood ratio scatter plot, showing bijective relationship between the two. (Right) Maximum-likelihood estimate obtained from the neural test statistic $\hat{\theta}_{\mathrm{ML}}$, compared with the true value $\hat{\theta}_{\mathrm{pLR}}$.

**Results**  As the ground truth is available in this model, we can compare the performance of the learned test statistic directly. In Fig. 3 (left) we show the ground truth profile likelihood ratio $t^{\mathrm{pLR}}$ as a function of the parameter of interest $\theta$ for 1000 datasets $\{x\} \sim p(\{x\} \mid \theta, \theta_\nu)$ with average cardinality of 110 events. The profile likelihoods have the typical parabolic structure with its minimum at the maximum-likelihood estimate $\hat{\theta}_{\mathrm{pLR}} = \mathrm{argmin}_\theta\, t^{\mathrm{pLR}}_{\{x\}}(\theta)$. The neural test statistic $t^{\mathrm{ML}}$ exhibits similar features, however as it is an output of a classification task is constrained to the interval [0,1]. The relationship between neural and profile likelihood ratio test statistic is shown in Fig. 3 (center), where a clear bijective mapping is recognizable. The network therefore learns an equivalent test statistic in terms of inferential power. Similar to the ground truth case, the best fit parameter can be found by minimizing the test statistic $\hat{\theta}_{\mathrm{ML}} = \mathrm{argmin}_\theta\, t^{\mathrm{ML}}_{\{x\}}(\theta)$ to produce a fully likelihood-free point estimate. In Fig. 3 (right) the two point estimates can be seen to correspond well to each other.

One of the major advantages of the neural test statistic is its inference time: While explicit profiling based on either a known or learned test statistic $L_x(\theta, \theta_\nu)$ requires multiple optimizations for each data instance, the neural network statistic allows for a fast vectorized computation in a single forward pass. We observe the neural network based inference for 1000 datasets to be over two orders of magnitude faster than the standard approach, i.e. explicit construction of the profile likelihood ratio computation of $\hat{\theta}, \hat{\theta}_\nu, \hat{\hat{\theta}}_\nu$ via optimization.

### 5.3  Mixture models in particle physics: Bayesian inference for a narrow resonance

**The forward model**  We also demonstrate a Bayesian dataset-wide inference for the mixture model case. Here the parameters of the model affect the mean of the signal component and the mixture coefficient, whereas the signal width and all background parameters are fixed hyperparameters. Using the notation in equation 8, we have

$$p_s(x_i \mid \theta_\nu) = \mathrm{No}(x_i \mid \theta_\nu, \sigma_s); \;\; p_b(x_i) = \mathrm{No}(x_i \mid \mu_b, \sigma_b); \;\; c_s(\theta) = \theta, \tag{10}$$

i.e. the parameter of interest $\theta$ is the relative signal strength and the location of the signal in the observed feature space is a nuisance parameter. In typical applications in particle physics, the signal is often very narrow, whereas the background distribution is more diffuse. All parameter choices and the priors $p(\theta)$ and $p(\theta_\nu)$ are specified in App. A. Instead of parameterizing the overall Poisson rate $\lambda$, here we used a fixed batch of events of size $N_0$, corresponding to the situation one might encounter in a real-time inference task. In summary, the dataset-wide joint probability is:

$$p(\{x\} \mid \theta, \theta_\nu)p(\theta, \theta_\nu) = p(\theta)p(\theta_\nu)\prod_{i=1}^{N_0}\left[c_s(\theta)\, p_s(x_i \mid \theta_\nu) + (1 - c_s(\theta))\, p_b(x_i)\right]. \tag{11}$$

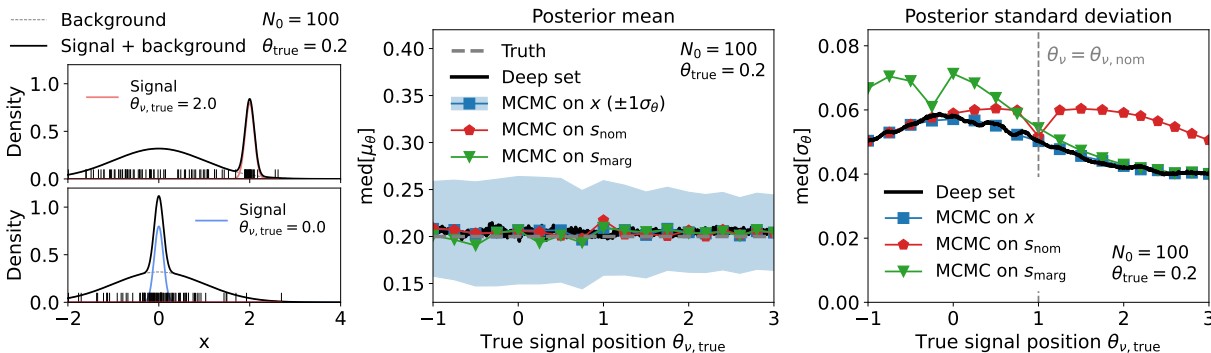

Figure 4: Signal- and background densities $p_s$ and $p_b$ in a simplified particle physics analysis for two different true values of the nuisance parameter $\theta_\nu$ (left), additionally showing sample datasets $\{x\}$ of size $N_0 = 100$. The median mean $\mu_\theta$ (center) and median standard deviation $\sigma_\theta$ (right) of the posterior $p(\theta \mid \{x\})$, obtained with different inference methods for different true nuisance parameter values $\theta_{\nu,\text{true}}$. The median is computed on ensembles of 400 datasets for each parameter point.

**Inference**  We wish to extract the posterior $p(\theta \mid \{x\})$ characterizing the prevalence of the signal process given a set of $N_0$ events, marginalizing over $\theta_\nu$. The mixture fraction $\theta$ is a fundamental property of nature; tracing its evolution over time is thus an excellent indicator of the health of the experimental apparatus. We use a deep set network to perform fast, amortized inference in this setting, trained as described in Sec. 4 directly on the set of observations $\{x\}$. Using the likelihood of equation 11, the posterior is also accessible through MCMC inference (as described in App. A), which can serve as a point of comparison. In many cases, the individual events $x$ are too high-dimensional for direct inference to be possible. A fundamental problem in the analysis of collider data thus consists in the construction of a low-dimensional summary statistic $x \to s(x)$, designed to retain the relevant information contained in the data, subsequently used in the inference step. It is common practice in contemporary work to use the signal-to-background density ratio $s_{\text{nom}}(x; \theta_{\nu,\text{nom}}) = p_s(x \mid \theta_{\nu,\text{nom}})/p_b(x)$ evaluated at a particular "nominal" nuisance parameter value $\theta_{\nu,\text{nom}}$. This observable is known to be a sufficient statistic for $\theta$ under the hypothesis $\theta_\nu = \theta_{\nu,\text{nom}}$ (Neyman & ES, 1933). Another popular choice is to neglect the hierarchical structure of the data-generating process; in the Bayesian paradigm this involves marginalizing over the nuisance parameters at a per-event level and using $s_{\text{marg}}(x) = \int d\theta_\nu \, p_s(x \mid \theta_\nu) \, p(\theta_\nu)/p_b(x)$ as summary statistic. Both quantities are known to not be information-preserving in general. Our example admits analytic likelihoods also for $s_{\text{nom}}$ and $s_{\text{marg}}$, allowing their performance to be evaluated through the MCMC procedure described above.

**Results**  The center and right panes in Fig. 4 compare the neural posterior estimate (black lines) to MCMC posterior estimates based on $x$ (blue markers). The width of the neural posterior closely matches the MCMC results, demonstrating the ability of the deep set to extract the entire information about the parameter $\theta$ contained in the data. Also shown are MCMC posterior estimates for $s_{\text{nom}}$ (red markers), and $s_{\text{marg}}$ (green markers). As expected, these generate wider posteriors[1] and, hence, weaker parameter constraints, illustrating the importance of a proper treatment of global nuisance parameters.

Using the fully parameterized density ratio $p_s(x \mid \theta_\nu)/p_b(x)$ for hierarchy-aware inference is, as mentioned in Sec. 2.3, complicated by the fact that often many hundred nuisance parameters are present in contemporary particle physics problems (The ATLAS Collaboration, 2022; The CMS Collaboration, 2022). However, nuisance parameters inform the the deep set through their presence in the training dataset and do not require any explicit parameterization of the network, suggesting a much better scaling behavior.

While we do not explore this direction further here, we note that the event-embedding $s_\phi^{\text{glob}}(x)$ learned by the deep set also plays the role of an information-preserving summary statistic that can serve as input to traditional (non-amortized) inference pipelines.

---

[1]In our simple example, $s_{\text{marg}}(x)$ is related to $s_{\text{nom}}(x; \theta'_{\nu,\text{nom}})$ for $\theta'_{\nu,\text{nom}} \approx -0.3$, leading to a sharp improvement in sensitivity for this model point. This is an artifact that is not expected to occur in more complicated environments.

### 5.4 Astrophysics example: Strong gravitational lensing

We test our procedure for hierarchical simulation-based inference on a more complicated problem from the domain of astrophysics, where the likelihood associated with the forward model is intractable – the characterization of the distribution of dark matter clumps (*subhalos*) in galaxies using images of galaxies gravitationally lensed by the clumps as well as a larger smooth mass distribution. The model contains both per-event (i.e., local) as well as dataset-wide (i.e., global) parameters. Estimating global lensing parameters with more sophisticated forward models was studied in, e.g., Anau Montel et al. (2023); Montel & Weniger (2022); Coogan et al. (2022); Wagner-Carena et al. (2022); Brehmer et al. (2019); we note that our aim here is instead to showcase the ability to do hierarchical simulation-based inference over local and global parameters with an amortized set-wide estimator.

$\longleftarrow$ Different latent realizations $z_\mathrm{sub} \sim p(z_\mathrm{sub} \mid \theta_\mathrm{glob}, \theta_\mathrm{loc})$ $\longrightarrow$

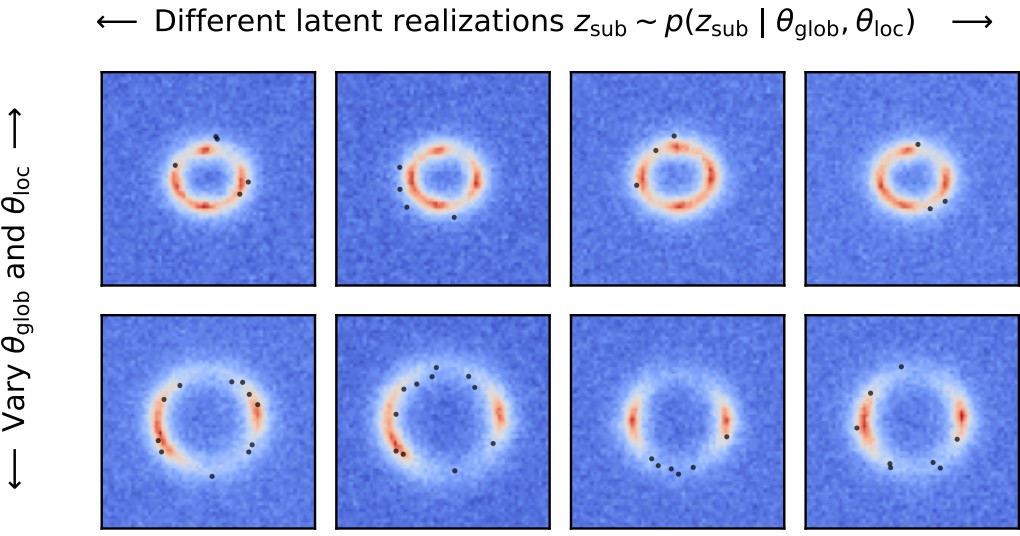

Figure 5: Illustrative samples from the lensing model. The rows show two different choices of local (per-event) parameters, while the different columns show variations on the global (set-wide) parameters for the fixed choice of local parameters. Sample-to-sample variation induced by the global parameters, which control the abundance of a subhalo population in the lens, can be seen. The scatter points shows the location of individual subhalos in each image.

**The forward model** The substructure mass function, which describes the mass distribution of clumps, is parameterized as a power law $\frac{\mathrm{d}n}{\mathrm{d}m_\mathrm{sub}} = \alpha \cdot m_\mathrm{sub}^{-\beta}$ where the normalization $\alpha$ parameterizes the overall abundance of subhalos, and $\beta$ is the spectral index. $\theta_\mathrm{glob} = \{\alpha, \beta\}$ are the global parameters in the hierarchical model. The number of subhalos follows a Poisson rate, $N_\mathrm{sub} \sim \mathrm{Pois}(\mu_\mathrm{sub})$, where we have the expected number of subhalos $\mu_\mathrm{sub} = \int \mathrm{d}m_\mathrm{sub} \frac{\mathrm{d}n}{\mathrm{d}m_\mathrm{sub}}$. In addition we have per-image (local) parameters $\theta_\mathrm{loc} = \{\theta_E, \Delta x, \Delta y, q\}$, where $\theta_E$ is the Einstein radius characterizing the overall scale of the lensing ring, $\{\Delta x, \Delta y\}$ is the offset vector between the centers of the background source and the lens galaxy, and $q \in [0, 1]$ is the axis ratio with 1 corresponding to a spherical lens. The subhalo radial positions are drawn from a uniform distribution close to the Einstein radius, $\{r_\mathrm{sub}\}_{i=1}^{N_\mathrm{sub}} \sim \mathrm{Unif}(\theta_E - 0.2'', \theta_E + 0.2'')$ and the azimuthal angle is drawn uniformly from the interval $[0, 2\pi]$. Given a realization of the masses and positions of the clumps $\{x_\mathrm{sub}, y_\mathrm{sub}, m_\mathrm{sub}\}_{i=1}^{N_\mathrm{sub}}$ (which otherwise act as latent variables, collectively denoted $\{z_\mathrm{sub}\}$), the expected image $\mu_\mathrm{image}$ can be obtained using a gravitational lensing simulator (Brehmer et al., 2019). The actual image is then a Poisson realization of this expectation, $x \sim \mathrm{Pois}(\mu_\mathrm{image})$. We choose our simulated lensed images to be $64 \times 64$ pixels in size; see App. A for more details on the lensing forward model. The per-image likelihood is intractable because of the high dimensionality of the latent space.

$$p(x \mid \theta) = \sum_{N_\mathrm{sub}=1}^{\infty} \int \mathrm{d}^{N_\mathrm{sub}} \{z_\mathrm{sub}\} \, \mathrm{Pois}(x \mid \mu_\mathrm{image}) \, p(\mu_\mathrm{image} \mid \{z_\mathrm{sub}\}^{N_\mathrm{sub}}, \theta_\mathrm{loc}) \, \mathrm{Pois}(N_\mathrm{sub} \mid \mu_\mathrm{sub}). \tag{12}$$

Some illustrative samples from the lensing forward model are shown in Fig. 5, showing variation with local (per-event) and global (set-wide) parameters across rows, and different realizations of the subhalo latent parameters (mass and location) along columns. The scatter points show the location of individual sub-clumps in each image.

**Inference**  We would like to infer the posterior over the local as well as global parameters $p(\theta, z \mid \{x\})$ given an ensemble of images. As discussed in Sec. 2, building per-image posterior estimators ignores the joint influence of the global parameters over the entire dataset, potentially reducing the identifiability of the local parameters. If a combined posterior for the global parameters is desired, combining per-image estimators into a global posterior can also be error-prone as per-observation errors accumulate, especially if flexible density estimators (e.g., normalizing flows) are used, as we do here. We train a hierarchical neural posterior estimator with datasets containing up to 25 lensed images, using a ResNet-18 (He et al., 2016) CNN as the per-event feature extractor. App. A describes further details related to model training.

**Results**  Figure 6 shows exemplary posterior inference results on one of the local parameters, the Einstein radius $\theta_E$ (left) and one of the global parameters, the amplitude of the clump mass function $\alpha$ (right), on test datasets. The red lines show posteriors using the hierarchical estimator, while the blue dotted lines show results of using a per-image estimator. We can see that the posterior mass concentrates around the true value (vertical dashed) as more images are analyzed. On the other hand, the local parameter values are very similar to those obtained without simultaneously constraining dataset-wide global parameters. This is because, in this case, sub-clumps have a minimal (sub-percent level) effect on the lensing image, and the degeneracy between the local and global parameters is weak.

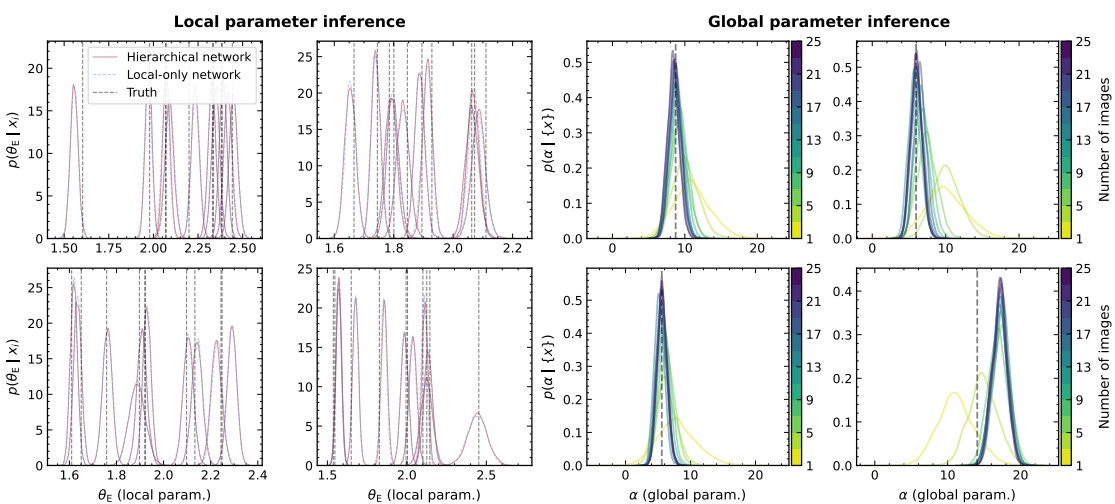

Figure 6: Example results of inference on local (left) as well as global (right) gravitational lensing parameters using the hierarchical simulation-based inference model, with the posterior on global parameter shown for various cardinalities as up to 25 images are analyzed. Concentration of the posterior mass around the true global parameter value (vertical dashed lines) is seen as more lensed images are analyzed, and the posterior distributions on the local parameters are seen to be qualitatively consistent with the true parameter values used for simulation.

**Statistical coverage**  We perform a test of posterior statistical coverage (described in detail in Hermans et al. (2021)) for the gravitational lensing example, serving as a validation check of the learned posterior estimator across cardinalities. Using the same notation as in the main text, let $\theta$ be the global parameter of interest and $\{x\}$ be a dataset. Denoting our learned posterior density estimator as $\hat{p}(\theta|\{x\})$. For a confidence level $1 - \alpha$, the expected coverage probability is

$$\mathbb{E}_{(\theta, \{x\}) \sim p(\theta, \{x\})} \left[ \mathbb{1}_\Theta \left( \theta \in \Theta_{\hat{p}(\theta|\{x\})}(1 - \alpha) \right) \right], \tag{13}$$

where $\Theta_{\hat{p}(\theta|\{x\})}(1-\alpha)$ gives the $1-\alpha$ highest posterior density interval (HDPI) of the estimator $\hat{p}(\theta|\{x\})$ and $\mathbb{1}_\Theta()$ is an indicator function mapping samples that fall within the HDPI to one. Given $N$ sampled datasets from the joint distribution $(\theta^\star, \{x\}) \sim p(\theta, x)$, the empirical expected coverage for the posterior estimator $\hat{p}(\theta|\{x\})$ is

$$\frac{1}{N}\sum_{i=1}^{N}\mathbb{1}_\Theta\left(\theta^\star \in \Theta_{\hat{p}(\theta|\{x\})}(1-\alpha)\right). \tag{14}$$

The nominal expected coverage is the expected coverage when $\hat{p}(\theta|\{x\}) = p(\theta|\{x\})$ and is equal to the confidence level $1-\alpha$. In Fig. 7 we show the results of a test of statistical coverage for the gravitational lensing example, showing empirical against nominal coverage probability for the same posterior estimator evaluated over different cardinalities (corresponding to the different lines shown), using 100 test samples. If an estimator produces perfectly calibrated posteriors, its empirical expected coverage probability and nominal expected coverage probability match (dashed diagonal line). The amortized posteriors obtained are shown to be well-calibrated across different cardinalities.

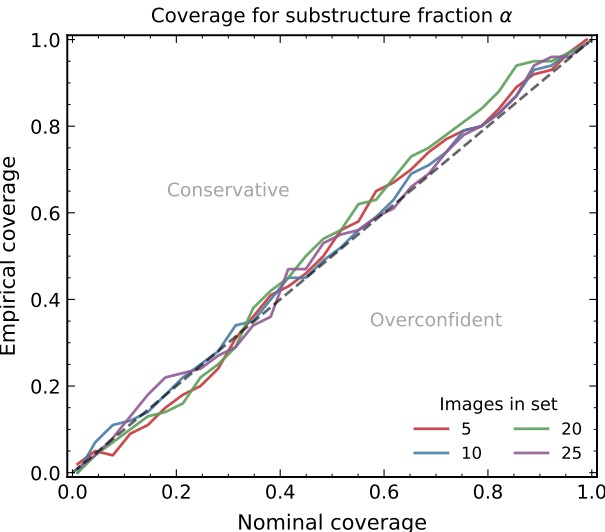

Figure 7: Test of statistical coverage for the gravitational lensing example, showing empirical against nominal coverage probability. If an estimator produces perfectly calibrated posteriors, its empirical expected coverage probability and nominal expected coverage probability match (dashed diagonal line). The amortized posteriors obtained are shown to be well-calibrated across different cardinalities (corresponding to the different lines).

## 6 Conclusions

We studied the problem of simulation-based inference over event ensembles in the presence of hierarchical structures in the underlying forward model. In this setting, common in the physical sciences, hierarchy-aware parameter inference requires a dataset-wide approach. For this purpose, we introduced neural estimators for likelihood-ratios and posteriors, illustrated their use for Bayesian as well as frequentist analysis in situations typical of astro- and particle physics, and substantiated correct inference performance in these scenarios. Our methods do not require explicit parameterization in terms of marginalized or profiled nuisance parameters, making them complementary to existing techniques for dataset-wide inference. Once trained, they provide fully amortized inference over datasets of varying cardinality and offer significant speedups over traditional frequentist and Bayesian methods, making them suitable also for high-throughput applications.

**Limitations** Our methods apply to situations where the cardinality $N$ of the observed dataset follows an arbitrary, but fixed, distribution $p(N \mid \theta)$ (cf. equation 2). It is currently an open question to what extent estimators trained on $p_1(N \mid \theta)$ achieve statistically sound performance when applied to datasets

corresponding to a different distribution $p_2(N \mid \theta)$, encoding, for example, the accumulation of additional data. We leave this extension for future investigation, but note that insights from recent work Berman et al. (2022); Chen et al. (2019) describing Bayesian updating as a dynamical system may offer a promising way to address this important problem.

## Acknowledgments

We made extensive use of the `Einops` (Rogozhnikov, 2022), `Jax` (Bradbury et al., 2018), `Jupyter` (Kluyver et al., 2016), `Matplotlib` (Hunter, 2007), `nflows` (Durkan et al., 2020), `Numpy` (Harris et al., 2020), `PyMC5` (Salvatier et al., 2016), `PyTorch` (Paszke et al., 2019), `PyTorch Lightning` (Falcon et al., 2020), and `Scipy` (Virtanen et al., 2020) packages.

LH is supported by the Excellence Cluster ORIGINS, which is funded by the Deutsche Forschungsgemeinschaft (DFG, German Research Foundation) under Germany's Excellence Strategy – EXC-2094-390783311. SM is supported by the National Science Foundation under Cooperative Agreement PHY-2019786 (The NSF AI Institute for Artificial Intelligence and Fundamental Interactions, http://iaifi.org/). This material is based upon work supported by the U.S. Department of Energy, Office of Science, Office of High Energy Physics of U.S. Department of Energy under grant Contract Number DE-SC0012567. CP acknowledges support through STFC consolidated grant ST/W000571/1. PW is supported by the Grainger Fellowship at the University of Chicago. This work was initiated at the Aspen Center for Physics, which is supported by National Science Foundation grant PHY-1607611. We thank the Munich Institute for Astro-, Particle and BioPhysics (MIAPbP) for their hospitality during the completion of this project.

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

# A  Additional details on experiments

## A.1  Details of 'Simple multi-variate normal likelihood' experiments

The deep set as well as transformer multivariate normal posterior estimators are trained on 50,000 sequences $\{x\}$ with a batch size of 128, withholding 10% of the samples for validation. Sequences consist of up to 200 elements, with each element having 15 features (3 multivariate normal dimensions with 5 draws per dimension). The means across the 3 dimensions of the multivariate normal are the parameters of interest $\theta$, and we target a posterior distribution over these.

A 3-layer MLP with RELU activations and hidden size 512 is used as the per-event encoder $s_\phi^{\text{glob}}$, which generates 256-dimensional per-event embeddings which are either aggregated (in the deep set model) or fed through transformer layers. An analogous decoder or prediction network $g_\psi$ then outputs the means and log-standard deviations, which are used along with the ground truth values to train the Gaussian density estimator $q_\varphi(\theta \mid \{x\})$.

The estimators are trained using the `AdamW` (Loshchilov & Hutter, 2019; Kingma & Ba, 2014) optimizer with initial learning rate $3 \times 10^{-4}$ and cosine annealing over 100 epochs. The checkpoint corresponding to the lowest validation loss is used. Evaluation is performed on 500 new test samples, evaluating the posterior estimator for sequences with cardinality varied from 1 to 200 for each test sequence.

### A.2 Details of 'Mixture models in particle physics: frequentist treatment' experiments

**Hyperparameters** The hyperparameters are chosen such that at the nominal parameter values $\theta = \theta_\nu = 1$ the total number of expected background events is $n_b = 100$ events, and the total number of expected signal events is $n_s = 10$. The hyperparameters of the Gaussian mixture components are $\mu_s = -7, \sigma_s = 2$ and $\mu_b = 0, \sigma_b = 3$.

**Minibatch sampling** The training algorithm for learning the test statistic for frequentist use requires choosing a weighting function in the $(\theta, \theta_\nu)$-space. For each minibatch the $(y = 0)$-labeled instances are sampled from the simulator $\{x\} \sim p(\{x\}|\theta, \theta_\nu)$ at $(\theta = \theta_0, \theta_\nu = \theta_{\nu,0})$. The parameter of interest is chosen randomly from a uniform distribution, $\theta \sim \mathrm{Unif}(3, 7)$, and a random nuisance parameter value is sampled from $\theta_\nu \sim \mathrm{Unif}(0.5, 2.0)$. The $(y = 1)$-labeled instances are sampled from $(\theta = \theta_0 \pm \Delta, \theta_\nu = \theta_{\nu,\pm})$, where $\Delta \sim \mathrm{Unif}(0.5, 2.0)$ and $\theta_{\nu,\pm} \sim \mathrm{Unif}(0.5, 2.0)$. Overall, each minibatch has $N = 100$ positive instances and $N = 100$ negative instances and the former are split equally between the two $\theta = \theta_0 \pm \Delta$ cases.

**Training** For each minibatch sampled for $\theta = \theta_0$, the instances $\{x\}_i$ are evaluated with the deep set conditioning parameter set to $\theta = \theta_0$ and the loss is evaluated as a standard binary cross-entropy loss averaged over the minibatch. The network parameters are optimized with the `Adam` optimizer with a learning rate of $\lambda_{\mathrm{elem}} = 10^{-4}$ for the per-element network parameters and $\lambda_{\mathrm{set-wide}} = 10^{-3}$ for the set-wide network parameters. Training proceeds for 30000 steps.

### A.3 Details of 'Mixture models in particle physics: Bayesian inference for a narrow resonance' experiments

**Choice of distributions and priors** The model hyperparameters are chosen to be $\sigma_s = 0.1$ for the signal and $\mu_b = 0$, $\sigma_b = 1$ for the background. The mixture coefficient $\theta$ is subject to a uniform prior, $\theta \sim \mathrm{Unif}(0, 1)$, and the nuisance parameter $\theta_\nu$ has a Gaussian prior $p(\theta_\nu) = \mathrm{No}(\theta_\nu|\mu = 1, \sigma = 2)$.

**Training of the deep set** The networks implementing the per-event encoder $s_\phi^{\mathrm{glob}}$ and decoder $q_\varphi^{\mathrm{glob}}$ are comprised of two dense layers with 128 hidden units and GELU activation functions; the event encoding consists of 64 features. 500k example datasets are generated from the prior distributions over $\theta_\nu$ and $\theta$; these datasets are used to train for 300 epochs with the `AdamW` optimizer, a batch-size of 256 and an initial learning rate of $10^{-3}$. Cosine-annealing is used over 300 epochs.

The evaluation is performed on independent datasets not used during training.

**Markov Chain Monte Carlo** We use the implementation of the `NUTS` algorithm available in `PyMC5` Salvatier et al. (2016). For each inference run, two Markov Chains of 50k samples each are sampled. The first 12k samples in each chain are discarded and not used for the estimation of the posterior.

### A.4 Details of 'Astrophysics example: Strong gravitational lensing' experiments

We generate 5000 ensembles (drawn from the same global parameters $\theta_{\mathrm{glob}} = \{\alpha, \beta\}$), each containing 25 lensed images with varying local parameters $\theta_{\mathrm{loc}} = \{\theta_E, \Delta x, \Delta y, q\}$. We use a ResNet-18 He et al. (2016) convolutional neural network as the per-event encoder $s_\phi^{\mathrm{glob}}$ to extract a 128-dim embedding vector from each image; half of the features are concatenated with the true global parameters and used to condition masked autoregressive normalizing flows Papamakarios et al. (2017); Rezende & Mohamed (2015) $q_{\varphi'}^{\mathrm{loc}}$ characterizing the local parameters for each image. The other half of the features are averaged, concatenated with the (randomly-varying at training-time) cardinality of the image dataset, and passed through a 3-layer MLP decoder, subsequently conditioning a normalizing flow $q_\varphi^{\mathrm{glob}}$ modeling the global parameters of interest.

The experiment is trained with a batch size of 16 (each batch containing $16 \times 25 = 400$ images), using the `AdamW` optimizer with cosine-annealed learning rate starting at $3 \times 10^{-4}$ for up to 100 epochs with early stopping, using the model with the best validation loss for evaluation. The sum of global- and local-normalizing flow log-likelihoods is used as the optimization objective.

## B  Motivation for loss function for joint posterior inference

We motivate the loss function used in equation 7, which factorizes the joint posterior over the local and global parameters $z_i$ and $\theta$ into a global posterior given the set $\{x\}$ and a product of posteriors over local parameters additionally conditioned on the global parameters:

$$-\mathcal{L} = \log q_\varphi^{\text{glob}} \left( \theta \mid g_\psi \left( \sum_{i=1}^N s_\phi^{\text{glob}}(x_i) \right) \right) + \sum_{i=1}^N \log q_{\varphi'}^{\text{loc}} \left( z_i \mid s_\phi^{\text{loc}}(x_i), \theta \right). \tag{15}$$

Given a hierarchical model with global parameters $\theta$ and local parameters $z_i$, and associated observed data $\{x\}$, we aim to demonstrate the factorization

$$p(\theta, \{z_i\}|\{x\}) \propto p(\theta|\{x\}) \prod_i p(z_i|\theta, x_i). \tag{16}$$

We make two key assumptions, given the nature of common hierarchical models. *(a)* Given the observations $\{x\}$, the global parameters $\theta$ are conditionally independent of the local parameters $\{z_i\}$, which gives us

$$p(\theta, \{z_i\}|\{x\}) = p(\theta|\{x\}) \, p(\{z_i\}|\theta, \{x\}), \tag{17}$$

and *(b)* Each local parameter $z_i$ is conditionally independent of any other $z_j$ given $\theta$ and its respective observation $x_i$, which gives

$$p(\{z_i\}|\theta, \{x\}) = \prod_i p(z_i|\theta, x_i). \tag{18}$$

Substituting the result from *(b)* into the expression from *(a)*, we obtain our targeted factorized form:

$$p(\theta, \{z_i\}|\{x\}) = p(\theta|\{x\}) \prod_i p(z_i|\theta, x_i) \tag{19}$$

justifying the loss function used in the hierarchical model.

