# OpenReview forum: "Hierarchical Neural Simulation-Based Inference Over Event Ensembles"
_TMLR — Accepted by TMLR_

### Review · Reviewer_pDeX · 2023-11-06

**Summary Of Contributions:**

The paper introduces an SBI-based method for hierarchical inference of both local and global variables. The authors then evaluate the method on a range of experiments and show that the method is able to recover meaningful posteriors in such hierarchical settings.

**Audience:**

Yes

**Broader Impact Concerns:**

No concerns.

**Claims And Evidence:**

Yes

**Requested Changes:**

- What is meant with "optimal inference"? This is used in many parts of the
  paper and since I am not familiar with the specific usage of "optimal
  inferece" I was quite confused by it's meaning here, as SBI in general is an
  approximate method that does not provide the true posterior. The term is also
  not introduced in the background section. Please explain what exactly is meant
  with "optimal inference" and define it properly in the paper.
- The structure of the paper could be more clear and it is difficult to distinguish
  background from the proposed method. Maybe having a section 2 purely for the
  background, and a new section 3 for the method, would be an option?
- Why is section 4 a separate section, not an experiment? It does feel a bit out
  of place in the current form and I was unsure how to interpret its meaning.
- The experiments are all somewhat differently structured, even though the
  paragraph headers suggest similarity. For example, some experiments describe the actual
  training procedure, i.e. what data the model is trained on, others do not. And while this can be found in the appendix I believe, Section 3.2 does not link to to it, so by the content
  of the section itself all training information is missing.
- Visualizing not just one of the proposed methods would be helpful. This could be done either by
  having a second figure, or maybe a visualization that includes both methods
  and thereby also highlights their similarities and difference is possible.
- The usage of paragraph headings: I believe they should only be used if the
  title actually belongs to the paragraph itself, but in the paper the text
  following the paragraph title consists of multiple paragraphs. So I strongly
  suggest either merging the paragraphs into a single paragraph, or actually
  using section headings. I believe that the former would be great in the
  experiments section, as often text was split into multiple paragraphs even
  though it clearly belongs together (e.g. section 3.3 "Inference"), or if it
  does not then the second paragraph should get a separate heading.
  On the other hand having actual subsections could be helpful e.g. in the
  background section to make the structure clearer, and since TMLR does
  not have a strict page requirement I do not really see any downside (also keep
  in mind that this way the table of contents displayed by many PDF readers
  becomes more accurate).
- In many instances the paper writes "Eq. equation [X].". Please fix the
  doubling.
- Section 2.2, last paragraph: In the current form the comment on LSTMs feels
  very out of place. I believe one way to fix this would be to move the "Deep
  sets in figure 1" sentence into the first paragraph as then the overall
  message of this paragraph becomes more clear.
- On the transformer-based architecture: It is only used in the first experiment
  in section 3.1, not in any other, and in this first experiment its performance
  is worse than the deep sets-based mode. So I am wondering if having the
  transformer in the paper is actually a helpful improvement, or if it is rather
  an unneccessary part that distracts from the main messages?
- Section 3.1: $\mu_0, \Sigma_0$ should be introduced more formally, e.g. by explicitly
  writing (I presume) $p(\theta) = N(\mu_0, \Sigma_0)$.
- Section 3.3: What do you mean with "we used a fixed batch of events of size
  $N_0$, corresponding to the situation one might encounter in a real-time
  inference task". Why is this a more realistic scenario? How does this relate
  to "real-time"? This was confusing to me.
- Figure 5: What is meant with "the local parameters are correctly recovered"?
  Looking at the left-most curve in the top left plot, this does not seem like a
  true statement. Overall, "correctly recovered" seems like a difficult
  statement in the context of Bayesian inference as we obtain a distribution
  over parameters, so making this more precise would be good.

**Strengths And Weaknesses:**

## Strengths
The paper proposes a new method that seems well-motivated, and the experiments
indicate that the proposed method is indeed able to provide meaningful
posteriors over parameters in a range of hierarchical inference tasks. The
experiments also seem relevant and thereby the paper provides a good motivation
for the settings in which the method could be applied in.

## Weaknesses
The biggest weakness to me is the structure of the paper. I listed more details
under "Requested Changes" below, but overall the text can in some parts feel
very unstrucutred, e.g. due to the many very small paragraphs that could be one,
the usage of paragraph headings as section headings, the
seemingly-clear-yet-unclear structure in the experiments (e.g. where "inference"
can include "training", but might also not), or the very separate section 4.
And more broadly, the presentation made it difficult for me to properly
understand which parts of section 2 are background, and which parts are the
actual proposed method.

---

> ### Author Response · Authors · 2023-12-05
> **Response to Review (Part 1)**
>
> We thank the reviewer for their helpful comments and suggestions, which have improved the clarity of presentation of the paper.
>
> **Q: What is meant with "optimal inference"? This is used in many parts of the paper and since I am not familiar with the specific usage of "optimal inferece" I was quite confused by it's meaning here, as SBI in general is an approximate method that does not provide the true posterior. The term is also not introduced in the background section. Please explain what exactly is meant with "optimal inference" and define it properly in the paper.**
>
> We appreciate the opportunity to clarify the usage of “optimal inference”. Throughout, by optimal inference we mean building a still approximate posterior or likelihood-ratio estimator that nevertheless takes into account the full assumed structure of the forward model. In the hierarchical setting explored, this is especially relevant because it is possible to build an SBI-based estimator that, e.g., leverages event-level information without accounting for the hierarchical structure of the forward model (as commonly done). In the frequentist setting, the SBI estimator of the profile likelihood is “optimal” also in the sense that it is the test statistic that maximizes power over equally distant (in the space of parameters of interest) alternatives.
>
> We have now clarified this through a footnote on the first page of the updated paper.
>
> **Q: The structure of the paper could be more clear and it is difficult to distinguish background from the proposed method. Maybe having a section 2 purely for the background, and a new section 3 for the method, would be an option?**
>
> We now substantially reorganize and revise the first few sections of the paper to clearly delineate background work from the current method. In particular, we give a general background on hierarchical inference in Bayesian and frequentist settings in Sec. 2, and describe related work in this context in Sec. 3. We describe our method in the new Sec. 4, which is also expanded to be more self-contained.
>
> **Q: Why is section 4 a separate section, not an experiment? It does feel a bit out of place in the current form and I was unsure how to interpret its meaning.**
>
> Indeed, thank you for pointing this out. We have now subsumed it into the gravitational lensing section, as its main aim is to validate the quality of the trained estimator across cardinalities in that experiment.
>
> **Q: The experiments are all somewhat differently structured, even though the paragraph headers suggest similarity. For example, some experiments describe the actual training procedure, i.e. what data the model is trained on, others do not. And while this can be found in the appendix I believe, Section 3.2 does not link to to it, so by the content of the section itself all training information is missing.**
>
> We now harmonize the discussions of the forward models, inference steps, and results across the experiments, with changes indicated in red. We also now explicitly link to the appendix for additional details from the various Experiments sections.
>
> **Q: Visualizing not just one of the proposed methods would be helpful. This could be done either by having a second figure, or maybe a visualization that includes both methods and thereby also highlights their similarities and difference is possible.**
>
> We attempt to include both the Bayesian and frequentist approach in the visualization of the method, as one of the central messages of the paper is that we are able to perform hierarchical inference in both paradigms. Although the transformer-based approach could also be shown, as we discuss below motivated by the review comments, we instead focus on the deep sets one, which is also reflected in the Figure.
>
> **Q: The usage of paragraph headings: I believe they should only be used if the title actually belongs to the paragraph itself, but in the paper the text following the paragraph title consists of multiple paragraphs. So I strongly suggest either merging the paragraphs into a single paragraph, or actually using section headings. I believe that the former would be great in the experiments section, as often text was split into multiple paragraphs even though it clearly belongs together (e.g. section 3.3 "Inference"), or if it does not then the second paragraph should get a separate heading. On the other hand having actual subsections could be helpful e.g. in the background section to make the structure clearer, and since TMLR does not have a strict page requirement I do not really see any downside (also keep in mind that this way the table of contents displayed by many PDF readers becomes more accurate).**
>
> We now confine paragraph headings to cases where it describes the full paragraph, collapsing paragraphs together as necessary in the Experiments section. As suggested, we also switch to subsections rather than paragraph headings in the background sections to delineate the different topics.

---

> ### Author Response · Authors · 2023-12-05
> **Response to Review (Part 2)**
>
> **Q: In many instances the paper writes "Eq. equation [X].". Please fix the doubling.**
>
> We have now fixed this in the paper.
>
> **Q: Section 2.2, last paragraph: In the current form the comment on LSTMs feels very out of place. I believe one way to fix this would be to move the "Deep sets in figure 1" sentence into the first paragraph as then the overall message of this paragraph becomes more clear.**
>
> As suggested, we have moved the Fig. 1 reference to the deep sets paragraph.
>
> **Q: On the transformer-based architecture: It is only used in the first experiment in section 3.1, not in any other, and in this first experiment its performance is worse than the deep sets-based mode. So I am wondering if having the transformer in the paper is actually a helpful improvement, or if it is rather an unneccessary part that distracts from the main messages?**
>
> Indeed, we found the transformer-based architecture to not perform as well as the simpler deep set-based one in our mult-variate normal experiment. We have changed some of the wording in the discussion surrounding the architecture, in particular mentioning it as an alternative to the set-based one with a perceived benefit (i.e., not requiring sampling of the cardinality at train-time), which is not realized in practice in the simpler experiment. We also now note in Sec. 4.1 that we use the deep sets approach for subsequent experiments.
>
> We nevertheless believe there is value in including a short mention and discussion of the autoregressive transformer as an alternative approach to varying-cardinality-set inference that has not previously (as far as we are aware) been explored in the literature. Overall, our main message does not hinge on the specific architecture – e.g., as noted in the background and related works, set-based aggregation towards simulation-based inference has previously been explored in the literature, and is not a novelty we highlight.
>
> **Q: Section 3.1: $\mu_0, \Sigma_0$ should be introduced more formally, e.g. by explicitly writing (I presume) $p(\theta)=N\left(\mu_0, \Sigma_0\right)$.**
>
> We now introduce $\mu_0, \Sigma_0$ more formally, in the form correctly suggested.
>
> **Q: Section 3.3: What do you mean with "we used a fixed batch of events of size, corresponding to the situation one might encounter in a real-time inference task". Why is this a more realistic scenario? How does this relate to "real-time"? This was confusing to me.**
>
> Using a fixed batch size for this example was simply a choice and not a limitation of the method. In particle-physics experiments, it is common for the low-level detector readout to be handled by digital logic implemented in hardware (on application-specific integrated circuits or programmable gate arrays) which implies very stringent resource constraints. Processing the incoming data stream in terms of fixed-size batches makes the resource usage deterministic and thus easier to integrate into the low-level hardware design as well as remain compliant with the latency budget.
>
> **Q: Figure 5: What is meant with "the local parameters are correctly recovered"? Looking at the left-most curve in the top left plot, this does not seem like a true statement. Overall, "correctly recovered" seems like a difficult statement in the context of Bayesian inference as we obtain a distribution over parameters, so making this more precise would be good.**
>
> Thank you for pointing this out – the language here was sloppy, and we have modified the caption to say, “posterior distributions on the local parameters are seen to be qualitatively consistent with the true parameter values used for simulation”.

---

### Review · Reviewer_sC5G · 2023-11-20

**Summary Of Contributions:**

The authors propose an amortized inference procedure for hierarchical models, It defines an inference network for individual observations, conditional on the global parameter, and a permutation-invariant global inference network. The posterior is factorized as the marginal posterior of the global parameter given all observations, and individual local latent variable posteriors conditional on the global parameter and the single local observation. The variational objective has a global term and a sum of local terms. The papers includes experiments on toy models, and various real models used in physics.

I'm having a hard time understanding what exactly are the claims of this paper and how the experiments support them.

**Audience:**

No

**Broader Impact Concerns:**

No concerns

**Claims And Evidence:**

No

**Requested Changes:**

Could the authors please point out specifically what are the falsifiable claims in the paper and how their experiments support those?

**Strengths And Weaknesses:**

The setup and method are quite clear, and appear sound to me. The experiments clearly list both the forward model and the inference procedure.

The method doesn't appear particularly novel, and I'm not really sure what the claims are. Below I comment on the three bullet points listed at the end of the introduction.
1. "We substantiate theoretically as well as empirically the fact that optimal inference in many implicit models with a hierarchical structure requires a dataset-wide approach, contrasted with the more common paradigm of combining implicit likelihood or posterior estimators associated with individual observations. We systematically derive conditions under which such a hierarchical approach is necessary for optimal inference, depending on how the model parameters are partitioned into local and global parameters of interest and nuisance parameters"
If the authors mean that the variational objective for the global parameter does not factorize over the observations in the presence of global nuisance variables, I would consider this well-known. If there's more to it, I can't find this analysis in the paper. I assume that "optimal inference" is some frequentist term of art, rather than a claim that this inference is somehow the best.
2. "We introduce frequentist as well as Bayesian methods for dataset-wide learning that can be used to perform simulation-based inference over event ensembles and can deal with datasets of varying cardinality. We connect our insights to several common use cases in the physical sciences and show how popular simulation-based inference paradigms can be adapted for optimal dataset-wide learning. In particular, we introduce the first approach for end-to-end frequentist simulation-based inference targeting hierarchical set-valued data"
I don't see a falsifiable statement here, that could be supported by experiments. As for novelty, amortized inference for datasets of unbounded cardinality has been studied before, such as by Le at. al. [1]. I can't comment on the novelty of the frequentist approach.
3. "We show that our machine learning-based inference methods are generically substantially faster than traditional approaches, such as Markov Chain Monte Carlo (MCMC) methods, even when the likelihood is tractable, while giving consistent results. They also allow performing inference in “streaming” mode where, e.g., posterior estimates are efficiently updated in real-time as new observations are made without having to perform a re-analysis of the entire updated dataset."
That variational methods are generally faster than MCMC is true, but hardly surprising. If history serves me well, that was the original motivation for studying them. The proposed method can be applied in a streaming setting, but I don't see any experiments showing that it performs well in such circumstances.

Additionally, the authors use a set encoder architecture as their main method, and a transformer network as a baseline which factorizes the global variational objective over the observations. The writing suggests that somehow this is a result of using a transformer, but I don't see why there couldn't be a transformer inference network using a global objective that does not factorize.

---

> ### Author Response · Authors · 2023-12-05
> **Response to Review**
>
> **Q: Could the authors please point out specifically what are the falsifiable claims in the paper and how their experiments support those?**
>
> We more sharply discussed our contributions in the updated paper. The main goals of this paper are twofold:
> - To describe frequentist as well as Bayesian methods that _(1)_ are sensitive to the full hierarchical structure of the forward model -- i.e., takes a dataset-wide view, treating local and global parameters separately, _(2)_ distinguish between nuisance parameters and parameters of interest at various hierarchy levels, and _(3)_ can process datasets of varying cardinality.
> - To showcase applications in scientific domains in order to empirically validate the methods.
>
> As noted by the reviewer and in the related works, prior studies have tackled subsets of the requirements in the first point, but as far as we are aware no prior work has described and tested a method satisfying all three. This concerns, in particular, the frequentist approach: to our knowledge this is the first time a simulation-based inference method for set-based data has been demonstrated for frequentist inference. We also acknowledge that the paper draws from prior work with the goal of demonstrating applications to the sciences, noting that the Scope of TMLR explicitly includes _"accounts of applications of existing techniques that shed light on the strengths and weaknesses of the methods"_.
>
> A falsifiable claim is therefore that the method should perform suitably when deployed over an ensemble of events – this is perhaps most convincing in Figs. 5 and 6, where we show for a forward model with an intractable likelihood that the obtained posteriors on parameters at both _local_ and _global_ hierarchy levels behave as expected, and have good statistical coverage over a range of different cardinalities. Further, Fig. 3 supports our claim that a suitably high-capacity model is able to recover a test statistic that is bijectively related to the true profile likelihood and therefore enables dataset-wide frequentist inference.
>
> **Q: …If the authors mean that the variational objective for the global parameter does not factorize over the observations in the presence of global nuisance variables, I would consider this well-known. If there's more to it, I can't find this analysis in the paper. I assume that "optimal inference" is some frequentist term of art, rather than a claim that this inference is somehow the best.**
>
> While it may be well-known, it is very common in simulation-based inference (SBI) applications when an ensemble of events is considered to construct and then combine per-event estimators (e.g., see https://arxiv.org/abs/1909.02005, https://arxiv.org/abs/2203.00690, https://arxiv.org/abs/2101.07263). Since we specifically target amortized SBI for scientific models with hierarchical structure, we consider this to be a valuable point to make. We also clarify our usage of “optimal inference” in a footnote on Page 1.
>
> Furthermore, in the specific context of frequentist inference, “optimal inference” refers to the fact that the profile likelihood is an “optimal” test statistic in the sense that it maximizes power over equally distant (in the space of parameters of interest) alternatives.
>
> **Q: “[…] They also allow performing inference in “streaming” mode [...]" That variational methods are generally faster than MCMC is true, but hardly surprising. If history serves me well, that was the original motivation for studying them. The proposed method can be applied in a streaming setting, but I don't see any experiments showing that it performs well in such circumstances.**
>
> The reviewer is correct that SBI methods are well-known to be substantially faster than traditional inference methods, and that this advantage is naturally present in our proposed method. This speed-advantage exists both for the first-time analysis of an entire dataset, as well as for the update of dataset-wide inference results as the dataset is being extended by new observations.
>
> While well-known, this speed advantage results in gains that are very relevant and beneficial in practical applications of the technique. We mention in Section 5.2, for example, that “We observe the neural network based inference for 1000 datasets to be over two orders of magnitude faster than the standard approach [...]”. This is very valuable in practical applications where inference needs to be performed on many datasets at once. Speedups of a similar magnitude were obtained for the Bayesian example in Section 5.3, where the MCMC was performed in a parallelized way on a cluster, while it was possible to train the deep set and perform inference on a single low-end GPU.
>
> As mentioned above, since the scope of TMLR explicitly includes applications highlighting existing methods, stress-testing the dataset-wide paradigm in a practical domain setting further exposes its advantages as relevant to machine learning application-focused communities.

---

### Review · Reviewer_Rc6M · 2023-11-22

**Summary Of Contributions:**

This paper proposes a neural simulation based inference method for hierarchical probabilistic models over event ensembles. The authors demonstrate that probabilistic inference in many implicit models with a hierarchical structure requires a dataset-wide approach. They introduce neural estimators for likelihood-ratios and posteriors for Bayesian as well as frequentist analysis. With experiments on synthetic, astro- and particle physics experiments. inference methods are generically substantially faster than traditional approaches,

**Audience:**

Yes

**Broader Impact Concerns:**

This paper proposed neural simulation based inference methods for hierarchical probabilistic models over event ensembles. There are no obvious ethical implications.

**Claims And Evidence:**

Yes

**Requested Changes:**

- The paper heavily relies on previous work. It is hard to follow because the relevant details are not included in the paper. Please provide a self-sustaining explanation of the method in the Section 2 and 3.
- I could not follow the derivation of (4). For the third equality to hold, it requires $p(\{x\}) = \prod_i p(x_i)$. This does not hold due to the existence of a dataset-wide latent variable $\theta$. Marginalizing out $\theta$ from $p(\{x\}, \theta)$ will let different $x_i$ correlate with each other. Did I miss anything in the derivation?
- On Page 3, it says that “it is not possible to combine marginal per-event posteriors…” From the probabilistic formulation, it is clearly possible to marginalize $\theta_v$ out from the distribution. Could you provide more details about why it could not happen?
- For the synthetic experiment in Section 3.1, it is no clear from the text what is inferred from data. What are $\sigma_1$, $\sigma_2$ and $\sigma_3$?

**Strengths And Weaknesses:**

Strengths
- Well-motivated problems with clear use cases in particle physics and astrophysics.
- Good empirical results showing the proposed methods can maintain inference quality while being significantly faster via amortized inference.

Weaknesses
- The paper does not contain a self-sustaining explanation of the proposed methods. This makes it hard to follow for the readers who have not worked on this problem.
- There is a concern regarding the derivation of (4).

---

> ### Author Response · Authors · 2023-12-05
> **Response to Review**
>
> **Q: The paper heavily relies on previous work. It is hard to follow because the relevant details are not included in the paper. Please provide a self-sustaining explanation of the method in the Section 2 and 3.**
>
> We have reorganized the first few sections of the paper to clearly delineate background work from the current method. In particular, we give a general background on hierarchical (simulation-based) inference in Bayesian and frequentist settings in Sec. 2, and describe related work in this context in Sec. 3. We then describe our method in the new Sec. 4, which is expanded to be more self-contained.
>
> **Q: I could not follow the derivation of (4). For the third equality to hold, it requires $p(x)=\prod_i p(x_i)$. This does not hold due to the existence of a dataset-wide latent variable $\theta$ . Marginalizing out $\theta$ from $p(x, \theta)$ will let different $x_i$ correlate with each other. Did I miss anything in the derivation?**
>
> For a hierarchical model with globally-shared parameter \theta, individual events are _conditionally_ i.i.d., i.e. $p(\\{x\\} \mid \theta) = \prod_i p(x_i \mid \theta)$, which is necessary for the third equality to hold. Note that here we are not marginalizing over $\theta$, as (4) targets the posterior distribution $p(\theta\mid \\{x\\})$.
>
> Indeed, we note in the very next line that, as noted by the reviewer, such a factorization would not be possible if instead we had global _latent_ variables, denoted, $\theta_\nu$:
>
> > However in the general case such combinations do not hold; e.g., it is not possible to combine marginal per-event posteriors over _global_ nuisance parameters $p(\theta\mid x_i) = \int \mathrm d\theta_\nu  p(\theta,\theta_\nu\mid x_i)$ to construct the dataset-wide posterior  $p(\theta\mid \\{x\\})$, marginalized over global and local nuisance parameters.
>
> **Q: On Page 3, it says that “it is not possible to combine marginal per-event posteriors…” From the probabilistic formulation, it is clearly possible to marginalize $\theta_\nu$ out from the distribution. Could you provide more details about why it could not happen?**
>
> Certainly. As pointed out by the reviewer above, if we have global latent variables $\theta_\nu$, the factorization in the third equality of (4) no longer holds. Therefore, we can no longer obtain the posterior over global parameters _marginal_ over the global latent parameters, $p(\theta \mid\\{x\\})$, as the product over individual posteriors marginalized over those global latents, $p(\theta\mid x_i) = \int \mathrm d\theta_\nu  p(\theta,\theta_\nu\mid x_i)$. That is, $p(\theta \mid\\{x\\}) \neq \prod_i \int \mathrm d\theta_\nu  p(\theta,\theta_\nu\mid x_i)$.
>
> The intuition behind this is that the per-event posteriors are individually “pre-marginalized” over the global latents, so combining them will not be the same as doing the integral over $\theta_\nu$ for the dataset-wide likelihood. One of the overall goals of the paper is to clarify when these kinds of assumptions (common in the sciences) can lead to sub-optimal inference results, and thus require a dataset-wide approach.
>
> **Q: For the synthetic experiment in Section 3.1, it is no clear from the text what is inferred from data. What are $\sigma_1$, $\sigma_2$, and $\sigma_3$?**
>
> Thank you for pointing out this omission – $\sigma_1$, $\sigma_2$, and $\sigma_3$ are the diagonal entries of $\Sigma_\mathrm{post}$, which is the covariance of the multivariate normal posterior on the target parameters $\theta$. $\theta$ is the mean of the multivariate normal likelihood. Hence, we are comparing the width of the posterior with the known analytic value. We now describe the plotted parameters in the text.

---

### Author Response · Authors · 2023-12-05
**Global response and summary of changes**

We thank all reviewers for their constructive reviews and comments, which have improved the paper and its presentation. We have updated the manuscript accordingly, highlighting the changes in red text. In particular,
- We have substantially revised the first few sections of the paper, as suggested by multiple reviewers, clearly delineating related prior works from the discussed method and sharpening the discussion of specific contributions;
- We have included several clarifying comments based on reviewer feedback where this improved the readability of the paper; and
- We have ensured a more unified treatment of paragraph and subsection headings, in particular in the discussion of the experiments and background.

In addition, we address reviewers’ comments individually.

---

### Decision · Action_Editor_dkwf · 2024-01-29

**Recommendation:** Accept with minor revision

**Comment:**

The authors have engaged in a thoughtful conversation with the reviewers and have improved the presentation of their manuscript. The reviewers agree that the work is correct and that the improved presentation clarifies the authors' claims. The only remaining point is the choice of the term `optimal` in the abstract and introduction. It also feels to me that the authors recognize that this can be misconstrued, hence their footnote on the first page. Since readers of the abstract will not have access to this footnote (and my general dislike for footnotes to define such critical aspects of an idea), I am recommending that the authors come up with a small modification that replaces the word optimal directly with what they mean — the effect leading to the footnote becoming superfluous. The definition of what the authors mean by optimal should be what is directly written in the text. I would suggest terms along the lines of "full-covariance approximate inference" or "hierarchy-aware posterior structure" but naturally leave this decision with the authors. I look forward to receiving their updated manuscript.

**Audience:**

While the material presented in this manuscript did not directly resonate with the selected reviewers, I think that there is a legitimate subset of the TMLR community that would be interested in this work. The idea of directly handling the hierarchical nature of certain probabilistic models will be of interest to anyone who works in approximate inference, especially those focused on simulation-based inference. The application of working with event ensembles will interest those working in physics and potentially other application areas, such as potentially geology and the social sciences.

**Claims And Evidence:**

All three reviewers presented concerns around the structure of the manuscript, which made its claims challenging to evaluate. After the authors incorporated the reviewers' comments, the claims have been clarified and the reviewers agree to their correctness. The evaluation is largely empirical, with examples drawn from physics and astronomy.

---

> ### Author Response · Authors · 2024-02-09
>
> Thank you for the good news, and all the reviewers for engaging thoughtfully with the process! We have now uploaded a camera-ready version of the paper. As suggested, we have removed the term "optimal", including at the level of the abstract, and replaced it with the term "hierarchy-aware". We have also moved the qualification of the term from a footnote on the first page into the main text, recognizing that this is a critical aspect of the paper. Finally, we have added example images drawn from the astrophysics forward model (Fig. 5) in order to clarify this example a bit more.